# Review of Groundwater Withdrawal Estimation Methods

**Marco Antonio Meza-Gastelum** [1]**, José Rubén Campos-Gaytán** [1,]*****, **Jorge Ramírez-Hernández** [2]**,**
**Claudia Soledad Herrera-Oliva** [1]**, José Juan Villegas-León** [1] **and Alejandro Figueroa-Núñez** [3]

1   Facultad de Ingeniería, Arquitectura y Diseño, Universidad Autónoma de Baja California,
Ensenada 22860, Mexico
2   Instituto de Ingeniería, Universidad Autónoma de Baja California, Mexicali 21280, Mexico
3   Comisión Estatal de Servicios Públicos de Ensenada, Ensenada 22800, Mexico
*   Correspondence: rcampos@uabc.edu.mx

**Abstract:** The demand for groundwater resources in arid and semi-arid regions has increased due to their progressive use in agriculture, industry and domestic activities. Among the difficulties and uncertainties that arise when managing groundwater resources is the calculation of groundwater withdrawals (GWW). The objective of this research work is to review the existing literature on the methods developed to estimate GWW by providing a summary of the advances, limitations and opportunities that the different methods developed on this topic could offer by identifying, categorizing and synthesizing the studies with a focus on developing a systematic guide so that researchers and practitioners conducting GWW studies can be informed of the most popular techniques, and the authors' experiences in recent years. Therefore, a literature search was conducted in the EEE, Google Scholar, SCOPUS, SpringerLink, ScienceDirect, Taylor & Francis Group and Wiley-Blackwell databases, using the following keywords: Groundwater AND (Withdrawal OR Pumping OR Abstraction) AND (Prediction OR Estimation). Thirty-four journal articles published between 1970 and 2021 were chosen based on the selection criteria, characteristics and capabilities of the approaches used for evaluation in GWW extraction. We concluded that the different methods for groundwater pumping estimation that have been reviewed in this work have advantages and disadvantages in their application. Direct approaches are very old and are still working uncertainty in their application is presented with possible human errors or in the measurement system. On the other hand, indirect methods have evolved along with technological advances, which have brought significant improvements and accuracy to these approaches.

**Keywords:** groundwater withdrawals; pumping well; estimate; extraction methods





## 1. Introduction

Water scarcity is among the greatest challenges facing humanity today, a situation that is alarming considering the pressure exerted on water resources in recent decades, due to rapid population growth, intense agricultural and industrial activity, and the high demand for water supply [1]. Groundwater accounts for one-third of the world's water demand and supplies drinking water to a large part of the population; however, in many regions, it is subject to stress both in terms of quantity and quality [2]. Mainly in arid and semi-arid regions, aquifers are subject to stress, which can be defined as a situation in which demand is greater than supply and often leads to overexploitation of aquifers [3]. In terms of groundwater use, agriculture is the main cause of scarcity, accounting for almost 70% of all groundwater withdrawals, and in some developing countries up to 95% [4]. It has been estimated that approximately 98.7% of all freshwater is available as groundwater. This resource provides 42%, 36% and 27% of the world's water used for agriculture, human use and industrial production, respectively [5]. Despite their importance for freshwater supply, groundwater resources are often poorly monitored, making it difficult, and sometimes impossible, to develop a consistent picture of their availability [6]. As a consequence, many

aquifers, particularly in semi-arid and arid regions, are currently overexploited because withdrawals exceed the rate of recharge [7]. This causes groundwater resources to come under pressure due to a number of factors, such as salinization, a critical problem that can contaminate groundwater and affect soil fertility, vegetation and ecological conditions along the coastal zones [8]. In addition, in deltaic areas, groundwater depletion can lead to land subsidence [9]. To avoid this catastrophe, it is necessary to properly manage the exploitation and protection of groundwater [10,11]. Since, if we are able to understand the aquifer's response to pumping before any damage to the aquifer system occurs, undesirable situations could be avoided [12]. Groundwater discharge (GWW) is an extremely important, often under-considered, and generally neglected component of water balance models [13]. Despite these monitoring needs, the follow-up and control of GWW are tasks that are generally perceived negatively by farmers, mainly because of exposing their usage habits [14]. Therefore, an ideal procedure for estimating the volume of groundwater extraction must maintain an acceptable level of accuracy, low (economic) cost and significantly reduce measurement errors [15]. With this in mind, the objective of this systematic review is to gain insight into the methods that have been applied to determine groundwater resources exploitation, providing a summary of the advantages, limitations and opportunities that the different methods developed on this topic can offer, identifying, categorizing and synthesizing the studies with the aim of producing a systematic guide for researchers and practitioners conducting GWW studies. A review such as this will help researchers understand the advances in research, and identify the strengths and weaknesses of each technique, which can help shape the direction of future research in this area. Therefore, first, the methodology used is presented. Next, a brief description of the methods is presented, followed by a citation and review of the studies conducted in this regard. This is followed by the discussion and conclusions.

## 2. Methodology

### 2.1. Search Strategy

A bibliographic search was carried out in the EEE, Google Scholar, SCOPUS, Springer-Link, ScienceDirect, Taylor & Francis Group and Wiley-Blackwell databases using the following keywords: (Groundwater [Title/Abstract]) AND (Withdrawal OR Pumping OR Abstraction [Title/Abstract]) AND (Prediction OR Estimation [Title/Abstract]). Articles were related to: hydrology, earth sciences, water resources and hydrogeology. The choice of keywords was intended to be simple and, since "Withdrawal", "Pumping" and "Abstraction" are synonyms generally used to refer to water withdrawals, they were also included in the search, using the Boolean operator OR. This filtering structure was applied to ensure consistency of the search across reference sources. No temporal restriction was imposed. Finally, a duplicate elimination was performed.

### 2.2. Screening and Eligibility Results

A first screening of titles and abstracts was performed, selecting articles that actually reported the use of methods to estimate groundwater withdrawals. The first eligibility criterion was to keep articles in which water withdrawals were focused in the areas of hydrology, earth sciences, agriculture, water resources, and hydrogeology. A second aspect considered articles related to the environment were not included, because these are based on the environmental impact caused by overexploitation in aquifers, without addressing the methodologies for estimating withdrawals. Once the above aspects were defined, we proceeded to the selection of articles for the review of the full text, dividing them into two categories: direct and indirect approaches to estimate GWW. In this work, direct approaches are considered to be methods or processes in which the volume of aquifer exploitation is measured in each pumping equipment manually or automatically. Whereas indirect methods refer to the process in which, alternative methods are used to estimate the exploitation volume. To perform these processes, the approaches established

by the PRISMA (Preferred Reporting Items for Systematic Reviews) methodology were followed [16]. This systematic review is described in Figure 1.

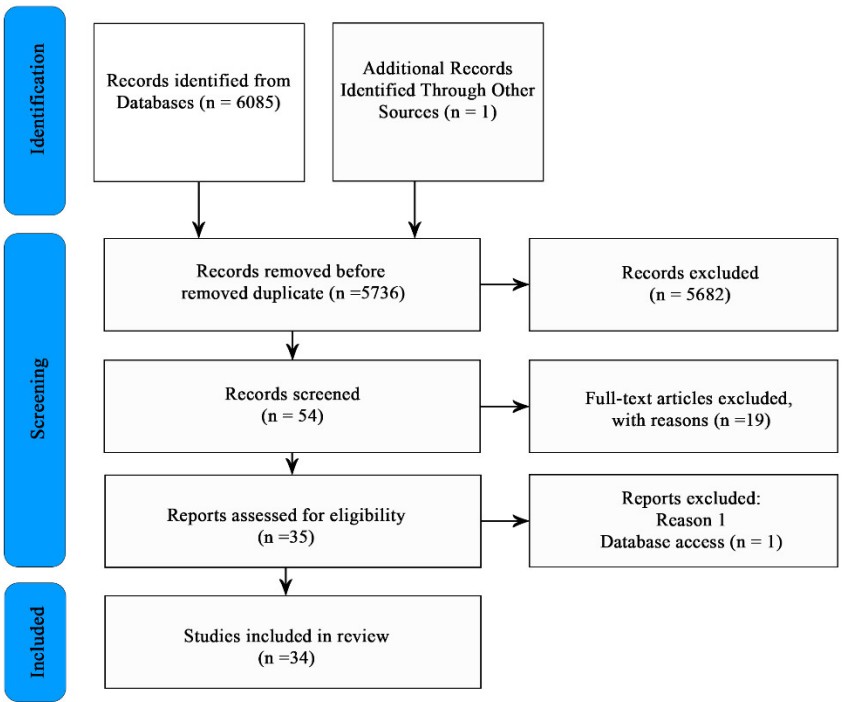

**Figure 1.** Process of search, selection and identification of studies according to the Preferred Reporting Items for Systematic Reviews and Meta-Analyses (PRISMA) 2020 flow diagram for new systematic reviews, which included searches of databases, registers and other sources [16].

### 3. Results

The databases consulted were EEE, Google Scholar, ScienceDirect, SCOPUS, Springer-Link, Taylor & Francis Group and Wiley-Blackwell, resulting in the identification of 30, 572, 1507, 119, 3429, 100 and 328 articles, respectively, with a total of 6085, in addition to 1 article included through snowball searching. Subsequently, duplicates were eliminated, leaving a total of 5736 articles. A first selection of titles and abstracts was then made, eliminating a total of 5682. The remaining 54 articles were exhaustively reviewed, discarding those that did not use techniques to estimate groundwater pumping, reaching a total of 35, of which 1 was discarded due to limited access to databases. This resulted in a total of 34 articles focused on quantifying GWW. Although no temporal restriction was imposed for the search, the first paper found with this criterion was from 1970.

Figure 2 shows the results of the articles analyzed by year. As can be seen, the highest number of publications is present in the year 2017 with four publications—during this year, direct and indirect methods received the same number of publications (two each)—followed by the year 2009 and the period 2018–2020 with three publications per year—during this period, the methods based on remote sensing and geographic information systems predominated. While the years 2010 and 2012 presented two publications per year focused on indirect methods. In addition, a constant of one publication can be observed for the period 1970–2005, a stage where direct methods and estimation method based on crop water demands were dominant; similarly, the period 2013–2015 only presented one publication per year—during this stage, models based on groundwater flow were the main approaches developed.

Figure 3 shows the publications made by each country. It can be seen that the United States is in first place with a total of 12 published articles, followed by China with 4 publications and in third place is Spain with a total of 3 publications. Despite not being present in other countries in Figure 3, this does not indicate that they do not have problems of

overexploitation; however, most of the research in this area is conducted in the United States and Asian countries with the highest depletion of groundwater worldwide (in terms of volume) [17]. Likewise, the importance of the subject can be observed for the United States, being the country with the highest number of publications and with the first studies carried out in the 1970s, denoting the importance of obtaining an adequate management system for the quantification of groundwater exploitation in their country.

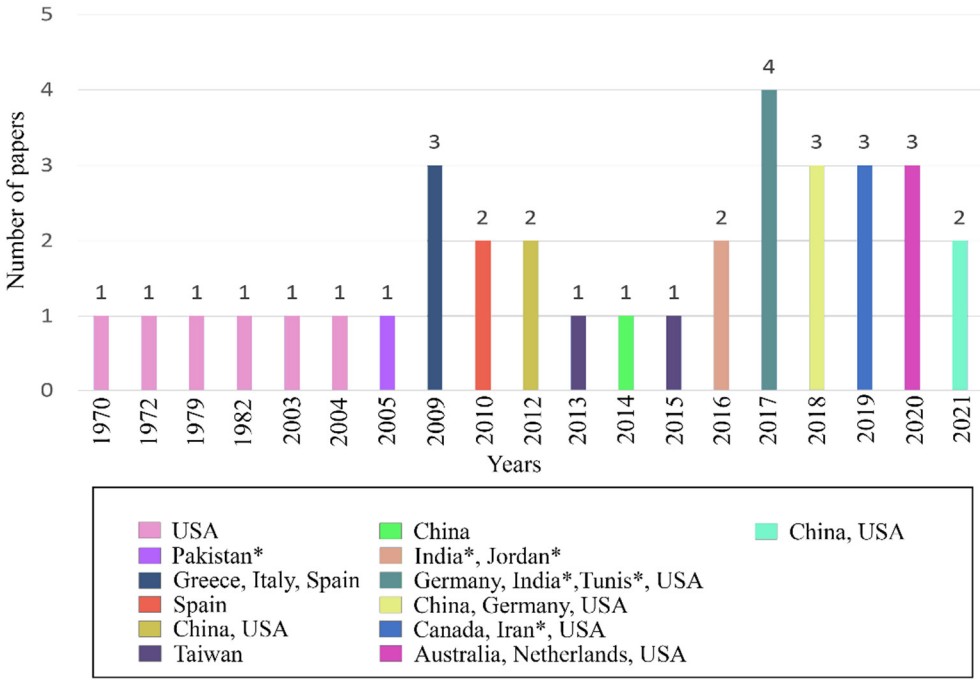

**Figure 2.** Annual publications of methods for estimating groundwater pumping. * Undeveloped countries.

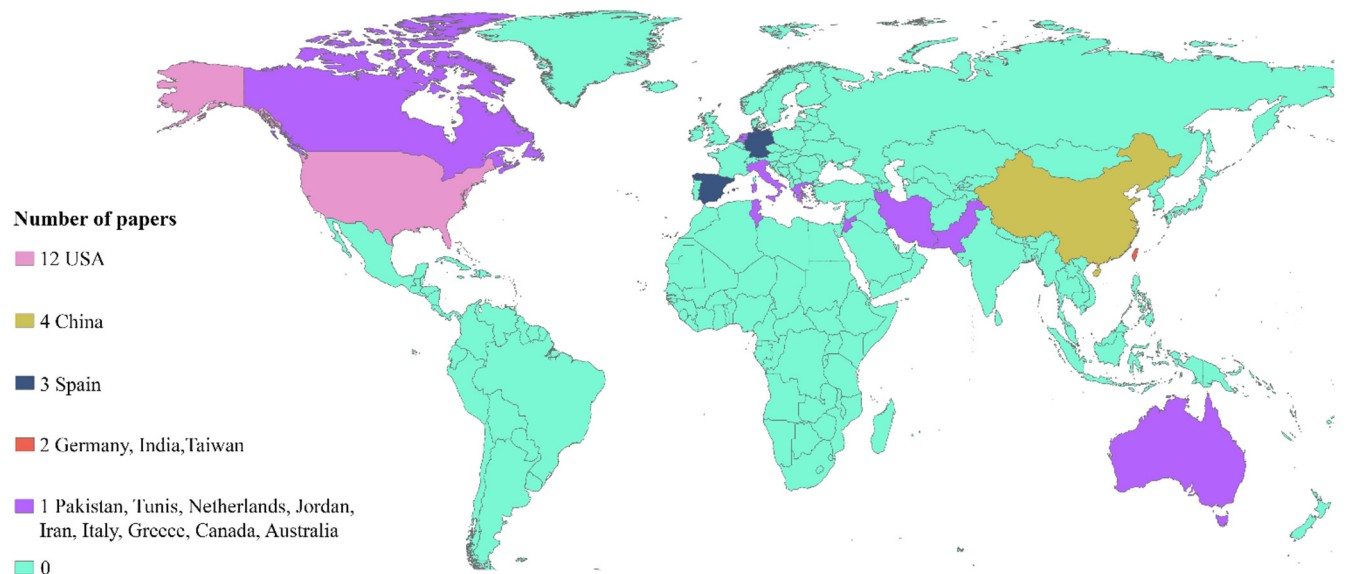

**Figure 3.** List of publications by country.

Finally, the investigations were classified into two categories according to the factors stipulated for this work: (1) association with direct methods; (2) association with indirect methods. These categories are described in Sections 3.1 and 3.2. At the end of this section, Table 1 shows a summary of the results obtained.

**Table 1.** Methodologies identified for pumping estimation as a function of the used approach. Root mean square (RMS); coefficient of determination ($R^2$); mean absolute error (MAE); root mean square error (RMSE); not specified (-).

| Study | Approach | Study Aquifer/Study Area | Number of Selected Pumping Wells | Scale of the Study | Accuracy of the Method or Characteristics | Method Classifiation |
|---|---|---|---|---|---|---|
| Martos-Rosillo et al. [18] | Time series analysis and numerical groundwater modeling | Lora and Mingo aquifers in Seville, Spain | 3 observation points | Area of smaller than 5 square kilometers | In the Lora aquifer, the mean annual extraction was estimated at $0.342 \times 10^6$ m$^3$, and that in the Mingo aquifer was estimated at $0.213 \times 10^6$ m$^3$. | Indirect |
| Forstner and Gleeson [19] | Multi-method sectoral approach | British Columbia, Canada | - | The average size of the aquifers varies, depending on the type, between 4 and 27 km$^2$ | Annual distribution of groundwater by sector: agriculture (38%), finfish aquaculture (21%), industry (16%), municipal water distribution systems (15%) and domestic users of private wells (11%). | Indirect |
| Vu et al. [20] | Two approaches: (1) local knowledge through a qualitative field study (of farmers), and (2) land use data combined with local knowledge on cropping and irrigation practices | La Vi River Basin, Vietnam | 77 wells | 100 km$^2$ | For the groundwater balance approach, an absolute error of $42.65 \times 10^6$ m$^3$ was calculated, which is equivalent to a relative error of 137% of the estimate. For the land use-based approach, the error estimate was $36.19 \times 10^6$ m$^3$, with a resulting range of $0$–$99.90 \times 10^6$ m$^3$. | Indirect |
| Tsanis et al. [21] | Combining surface-water and groundwater potentials using a conceptual rainfall–runoff model (the Sacramento hydrologic model) | Messara Valley, Crete, Greece | 11 climatological stations | 400 km$^2$ | The average error for the entire period 1981–2002 is 20%. | Indirect |
| Yang et al. [22] | Water table fluctuation regression (WTFR) method | Dagu aquifer, coastal area in eastern China | 82 wells | 430 km$^2$ | The RMS of water level in all wells ranged from 0.05 to 0.49 m, with a mean of 0.22 m (0.22 m). | Indirect |
| Bhadra et al. [23] | Three methods: (1) Discharge discharge factor (DF); (2) linear groundwater consumption models (LGDM), and (3) irrigation depth | 70 villages in Sikar district, Rajasthan state, northwest India | 6696 wells to determine the DF and 39 observation wells | 620 km$^2$ | The RMSE with the LGDM model in elevation zones: shallow, intermediate, deep and combined vary between 0.43 and 0.34. | Indirect |

**Table 1.** *Cont.*

| Study | Approach | Study Aquifer/Study Area | Number of Selected Pumping Wells | Scale of the Study | Accuracy of the Method or Characteristics | Method Classifiation |
|---|---|---|---|---|---|---|
| Casa et al. [24] | Based on crop needs, using GIS and remote sensing techniques in combination with water balance | Pontina Plain in Central Italy | - | 700 km$^2$ | The results of this study estimated a demand of 70 Mm$^3$ per year, i.e., 100 Mm$^3$ per year of irrigation needs if an average irrigation application efficiency of 70% is considered. | Indirect |
| Li et al. [25] | Two methods: (1) the water balance method, and (2) back-propagation artificial neural network (BPANN) | Tongzhou, southeast Beijing | 14 wells | 906 km$^2$ | The water balance method $R^2 = 0.9914$ and the BPANN method $R^2 = 0.9069$. | Indirect |
| Dubois et al. [26] | Approach based on water requirements for crops | Nebhana Plain, northeastern Tunisia | - | 1221 km$^2$ | Accuracy crop classification improves by 10% with multitemporal features. | Indirect |
| Ruud et al. [13] | Water balance model based on geographic information systems (GIS) | Southern San Joaquin Valley, California, USA | - | 2300 km$^2$ | The model estimate has an error of 38%. | Indirect |
| Lin et al. [27] | Groundwater equilibrium model with coupling surface-water (SWAT) and groundwater (MODFLOW) modeling | Multi-aquifer system in an alluvial fan of the Choushui River, Taiwan | The entire aquifer | 2500 km$^2$ | MAE = 5.1%. | Indirect |
| Liu et al. [28] | Two-part approach, Independent Component Analysis (ICA) and groundwater modeling | Aquifer in the Sijhou municipality, in the alluvial fan of the Jhuoshuei River, west-central Taiwan | 136 observation wells | 2562 km$^2$ | The calibration of the groundwater model had an RMSE of 1.00 m. The total amount of groundwater pumped is 37,565 m/day, of which more than 58% is for agriculture and 42% is for industrial use. | Indirect |
| Ray et al. [29] | Approach based on spatio-temporal patterns of groundwater consumption for irrigation | Seonath-Kharun, India | 43 wells | 2867 km$^2$ | Annual estimate of groundwater consumption for irrigation was $212 \times 10^6$ m$^3$. | Indirect |
| Baker [30] | Statistical approach, agricultural water uses and electrical measurements | Kansas, USA | 78 wells equipped with flowmeters | 1800 mi$^2$ | Statistical method: the standard deviation was 1.22; estimates of the variance for irrigated area of the total sample was 40.23 and pooled was 0.77. The estimates of the sample variance from energy consumption coefficients was 1.868. | Direct |

**Table 1.** *Cont.*

| Study | Approach | Study Aquifer/Study Area | Number of Selected Pumping Wells | Scale of the Study | Accuracy of the Method or Characteristics | Method Classifiation |
|---|---|---|---|---|---|---|
| Martínez-Santos and Martínez-Alfaro [31] | Coupling of the water table fluctuation (WTF) method with the groundwater balance equation | Western Mancha Aquifer, Spain | - | 5500 km$^2$ | Average error rate of 10%. | Indirect |
| Castaño et al. [32] | GIS-based method focused on the quantification of withdrawals irrigation | Mancha Oriental, Spain | - | 7260 km$^2$ | Accuracies greater than 95% with a cost 60-fold lower than traditional methods. | Indirect |
| Parizi et al. [33] | This approach called "representative pumping well network" (RPWN) | Three aquifers in Iran: Tehran, Arak and Qazvin | 50 pumping wells for each aquifer | Tehran aquifer with an area of 2250 km$^2$, Arak aquifer with an area of 1945.5 km$^2$ and Qazvin aquifer with an area of 3952.0 km$^2$ | RPWN shows errors between 0.2% and 1.41% with respect to actual RPWN. | Indirect |
| Su et al. [34] | Numerical model coupled with ModFlow2005: groundwater flow and land subsidence | Multi-aquifer system in Tianjin Plain, China | 136, 87, 53, 42, and 4 wells, respectively for each aquifer | 10,600 km$^2$ | The average annual exploitation based on the model was estimated of $8.35 \times 10^8$ m$^3$ per year. The average annual exploitation derived from the quota method and statistical data were $9.73 \times 10^8$ m$^3$ and $5.45 \times 10^8$ m$^3$ per year, respectively. | Indirect |
| Al-Bakri et al. [35] | GIS-based method focused on the quantification of irrigation withdrawals | Three basins in Jordan: Yarmouk, Amman-Zarqa and Azraq | Yarmouk 129 wells, Amman-Zarqa 590 and Azraq 488 wells. | The area of the Yarmouk basin is approximately 1393 km$^2$, the Amman-Zarqa basin has an area of 3600 km$^2$, the Azraq basin has an area of 11,742 km$^2$. | Irrigated crop maps showed good agreement between irrigation maps and soil data, with an overall accuracy of 87%. Groundwater over-extraction was estimated to be between 144% and 360% of the safe yield in the three basins. | Indirect |
| Moreo et al. [36] | Three-dimensional numerical and transient model | 34 study areas in Death Valley region (Nevada and eastern California), USA | 9300 wells | 19,000 mi$^2$ | Error in extractions estimation was 8.3%. | Indirect |

Table 1. *Cont.*

| Study | Approach | Study Aquifer/Study Area | Number of Selected Pumping Wells | Scale of the Study | Accuracy of the Method or Characteristics | Method Classifiation |
|---|---|---|---|---|---|---|
| Shao et al. [37] | Inversion method based on a numerical model | North China Plain (NCP) | 101 observation wells | 139,000 km$^2$ | In a synthetic case with an accuracy of 0.1 m, the average error during 10 years was 1.47% and 1.54% in two pumping subareas. Instead, with an accuracy of 0.01 the average error was 0.06% and 0.13%. The average estimate of the groundwater extraction for the NCP was of 24.92 $\times$ 10$^9$ m$^3$ per year. | Indirect |
| Tillman et al. [38] | Approach based on crop water requirements | Alluvial basins of southwestern Arizona, USA | - | 190,000 km$^2$ | The annual estimated volume of groundwater discharge by vegetation was between 1.4 and 1.9 million of m$^3$ per year, with an annual mean of 1.6 million of m$^3$. Correlation coefficients between monthly and annual groundwater discharge by vegetation and precipitation were low, (r = 0.182, $p > 0.05$) and ($p > 0.05$), respectively. | Indirect |
| Wray [39] | Approach based on water requirements for crops | High Plains, USA | - | 455,000 km$^2$ | Groundwater pumping was sampled in 15 counties in the region. | Indirect |
| Ahmad et al. [40] | GIS-based water balance model | Rechna Doab, Pakistan | - | Area of approximately 2.97 million hectares | The specific yield method produced 65% lower net groundwater use compared to specific yield. | Indirect |
| Ogilbee & Mitten [41] | Two methods: (1) electrical energy consumption, and (2) electrical consumption coefficient | Santa Clara County and Fresno County, California, USA | 200 wells | - | Electricity consumption 4%, 10% for electricity consumption coefficient. | Direct |
| Luckey [15] | Random sampling combined with regression analysis | Arkansas River Valley in southeastern Colorado, USA | 225 wells | - | Estimation of the standard error, 14% for random sampling and 10% for regression analysis. | Direct |
| Alfaro et al. [42] | Numerical model-based approach and crop water requirements | Eastern part of the Lower Jordan Valley southern part | 6 wells | - | Pearson correlation coefficient (R), calculated versus measured values of 0.90 and 0.97, for the calibration period. | Indirect |

Table 1. *Cont.*

| Study | Approach | Study Aquifer/Study Area | Number of Selected Pumping Wells | Scale of the Study | Accuracy of the Method or Characteristics | Method Classifiation |
|---|---|---|---|---|---|---|
| Harris and Diehl [43] | The USGS model is compared to two source data, the EIA (Energy Information Administration) and the USGS compilation | Groundwater-fired power plants in USA | 470 plants for the USGS compilation; 742 plants the EIA | - | Error percentage of the estimates was 17% for EIA and 24.6% for USGS compilation. | Direct |
| Massuel et al. [14] | Direct measurement methods and adaptive methodology that sought to involve the users | Three locations in North Africa (Morocco, Tunisia and Algeria) | Saïss 430 wells; Kairouan 928 wells, and Ziban 1255 wells | - | Water pumping in Saïss has a −20% of discrepancy compared with the developed method. The average annual pumping time in Kairouan was 4300 h/year corresponding to a calculated average annual volume of 71,000 m$^3$/year. At Ghrouss, the annual groundwater abstraction was estimated in 170 hm$^3$/year. | Direct |
| Majumdar et al. [44] | Holistic approach combining water balance components with a machine learning model | The High Plains Aquifer (HPA), located in the central United States | - | - | In a test case, the method predicted groundwater withdrawals with R$^2$ ≈ 0.23, MAE ≈ 16.01 mm, RMSE ≈ 31.51 mm and normalized MAE ≈ 0.84. | Indirect |
| Kent et al. [45] | Approach based on historical records and data hierarchy to establish the number of wells and their extraction characteristics | Extent of catchments associated with the Great Artesian Basin (GAB) and other regional-scale groundwater basins in Queensland, Australia | 96,174 drill holes | - | Current GAB groundwater use in Queensland was estimated at 322 GL/year. | Indirect |
| Martindill et al. [46] | Ratio-based approach between pump energy consumption and Efficiency Lift Method (ELM) | San Joaquin Valley, California, USA | 30 wells studied from 2010 to 2015 | - | For individual tests, MAE = 13.5%; per month, MAE = 5%. | Direct |

*3.1. Direct Approaches to Estimating GWW*

Methods in which the withdrawal volume at each pumping station is measured (manually or automatically) or calculated using the time-rate method by multiplying the average pump speed by the cumulative pumping time are known as direct approaches [47]. The most accurate and reliable method of monitoring GWW is to install flowmeters on all groundwater pumping wells [46]. However, flowmeters are generally not applied in all regions; moreover, owners of extraction wells are generally unwilling to install new flowmeters themselves that would allow accurate monitoring of groundwater withdrawals in agricultural wells [14,48]. Currently, groundwater-mining research continues to adopt direct methods for groundwater pumping estimation; however, the application of these methods can be difficult, as direct measurements require data that are often underreported which could pose a challenge for accurate estimates [34,49]. Despite this, estimation of pumping rates can be carried out by transforming electrical energy consumption data acquired from wells performing groundwater withdrawals [47]. Since the relationship between electricity consumption and pumping volume is a reliable relationship, the accuracy of these methods depends on the census of pumping wells coupled with complete and accurate records of pump electricity consumption, more accurate data lead to a more accurate estimation of pumping [28]. For example, Ogilbee and Mitten [41] estimated municipal and agricultural groundwater pumping for the major basins of California in the United States. They developed tables of total annual electricity and natural gas consumption data for groundwater pumping, which were obtained from the major utilities in central California. Data obtained correspond to the volume of groundwater pumped for municipal use in 27 communities ranging in population from 1000 to 145,000. The methodology for determining the annual per capita use factor, which is used to determine the volume of water supplied for municipal use, was done by dividing the volume of water by its population, so that the volume of water pumped can be calculated by multiplying the population of a community by the annual per capita use factor. While to estimate groundwater pumping for agricultural use, pumping as a function of surface area was used, using the power coefficient method. Meanwhile, Martindill et al. (2021) [46] focus on the relationship between the energy consumption produce of the pumps used in the GWW by employing the efficiency lifting method (ELM), due to the availability of electricity data and the operating conditions of the pumps have made the ELM method feasible to estimate the GWW on a large scale.

Furthermore, using this method provides a framework for converting well pumping energy into an estimate of GWW. Specifically, methods based on electrical measurements state that the total amount of pumping for each well is a product of total electricity consumption and pumping time [50,51].

Likewise, traditional statistical methods are often practical and applicable for estimating groundwater withdrawals [25]. For example, Luckey [15] proposes statistical methods to determine annual groundwater withdrawals in areas containing a high number of extraction wells, using regression analysis techniques or random sampling. This methodology was developed in the Arkansas River Valley, USA, where was used a random sample of 100 extraction wells in an area containing approximately 896 wells. Demonstrating that once the population type and the parameters for the year in which the random samples are collected are known, the annual groundwater withdrawals could be calculated using random sampling methods, being a solution applicable to any area with a large number of wells.

On the other hand, Baker Jr. [30] analyzed a small sample of well extraction values, using part of the statistical approach used by Luckey [15], which applied to western Kansas, USA, in an attempt to estimate groundwater extraction with acceptable accuracy over a large area. To evaluate such purpose, the following techniques were used: extrapolation of a groundwater withdrawals sample, crop requirements from rainfall and irrigated areas of various crops types, and withdrawals from a sample of power coefficients.

Harris and Diehl [43] compare three sets of Thermoelectric Water-Withdrawal data, one based on USGS compilation data [52], one based on Energy Information Administration (EIA) data, and one based on estimated USGS model data [53]. The purpose of this comparison is to better understand the uncertainties in the extraction data used in thermoelectric plants, so comparisons were made between subsets of plants with the same cooling system in pairs of datasets: USGS model vs. EIA, USGS model vs. USGS compilation, and EIA vs. USGS compilation. Quantifying the uncertainty in reported extraction data by defining agreed extraction and discrepant extraction. The minimum value is referred to as agreed extraction and is accounted for in each of the three datasets, whereas discrepant extraction is the calculated difference between the highest extraction rate and the agreed extraction rate that was present in all three datasets.

Additionally, seeking to involve users is a task that would facilitate the estimation of groundwater pumping; for example, Massuel et al. [14] did this with a user-oriented methodology, taking them as a starting point by involving them in the study in addition to the implementation of different types of censors adapted to the conditions of the well or pump used for water extraction. This methodology is supported by a process of shared knowledge through the understanding of the users, who are involved by sharing information through surveys related to the configuration of the irrigation system, the cropping pattern and cropping calendar among others. The interviews were based on personal history in the area, the users' concern about water resources, etc. In this way, the farmers' habits and irrigation practices are related. Likewise, relating pumping to electrical expenditure could present a deficiency in the application of this method since the results obtained in this work indicated that the evaluation of groundwater withdrawals was exposed to the constant changes made by farmers in their equipment and farming systems. Similarly, Kent et al. [45] used an approach based on historical records of groundwater use trends and data hierarchy to establish the number of wells and their withdrawal characteristics. The first part of the model consists of determining the wells that may be extracting groundwater. A two-part approach was used for this first stage: (1) using a hierarchical approach for existing borehole datasets, giving priority to the highest quality/trusted data; and (2) for boreholes or wells where this information is not available, a spatial approach was developed to assign the location of those boreholes or wells. Subsequently, the type of use was established to determine withdrawals according to the purpose of the well or borehole, establishing a different methodology according to the use sector, e.g., livestock supply and domestic uses, discharge from uncontrolled artesian boreholes, water extraction in sectors of oil and gas uses. Estimation of groundwater use for each purpose and aquifer group was presented visually on a 50 km × 50 km mosaic grid, with weighted shading based on extraction volume.

### 3.2. Indirect Approaches to GWW Estimation

In this work, the estimation of the pumped volume based on the effects produced in the aquifer by the water extracted from the itself aquifer, or by the application of the water to a specific use, is referred to as indirect approaches and could be classified into five groups according to the characteristics and tools of the approach: estimation method based on crop water demands; estimation methods based on satellite data; water table fluctuation (WTF) method; groundwater modeling approach; and artificial intelligence-based methods [33].

***Estimation method based on crop water demands***: Among the commonly used approaches is to measure the irrigated crop area and multiply it by the water requirements of each crop; this approach is used in aquifers where the main activity is based on agriculture [54,55]. Such as Wray [39], who developed a study to estimate the withdrawal water volume of using techniques that combine multispectral data with groundwater pumping data for different cropping seasons. This method defines the location and extent of the lands that have crops, some of which may have variations of such crops. To define the crop characteristics, a computer analysis of the Landsat satellite data is performed, so that the amount of water used is calculated by accumulating the product of the crop area by

the average pumping for the water needs of each type of crop. Another example of this application is Tillman et al. [38], who estimated groundwater discharge based on natural vegetation for an area of over 190,000 km$^2$ using satellite remote sensing data for a watershed in the southwestern of Arizona, USA. Enhanced Vegetation Index (EVI) data from EOS-1 MODIS sensors were used. In addition, a relationship between evapotranspiration, EVI and temperature was established to calculate groundwater discharge from vegetation. The objective of this project was primarily focused on developing a simple method to determine GWW using vegetation in an area where detailed information on soil moisture and water table depth is not available. Bhadra et al. [23] evaluate three approaches—the first one is the discharge factor (DF), which is based on the groundwater consumption volume through wells per unit of crop area. The second approach was developed using linear groundwater consumption models (LGDM), which is based on the empirical relationship between satellite-derived crop area and groundwater consumption with respect to crop demand. The third approach was based on irrigated area, which is estimated by a sample survey of the different crops, water requirements and irrigation method. Of the three methods the LGDM proved to be the most realistic, demonstrating a good application of methods using satellite data in arid regions. Likewise, Dubois et al. [26] determine the water need in agricultural areas, where groundwater consumption is evaluated based on optical data obtained from the Sentinel-2 satellite for each season (summer or winter). The process consists of determining the Normalized Difference Vegetation Index (NVDI). NDVI profiles are created for the different types of crops, whose profiles are used to evaluate the crop surface with greater precision in different seasons of the year. It is possible to estimate the water consumption according to the crop needs and thus estimate the groundwater discharge by multiplying the crop surface by the corresponding water consumption of the vegetation. On the other hand, Casa et al. [24] carried out an estimation of crop water requirements in the Pontine plain, in central Italy, using remote sensing and the application of water balance in a GIS environment. The methodology, based on remote sensing and GIS, used four Landsat ETM+ images and meteorological and geographical vector layers. The study seeks to determine the water needs according to the type of crop, using a methodology based on crop evapotranspiration, monthly precipitation values and the water available in the soil of the plants. For this purpose, a hypothetical evapotranspiration mapping of the study area was developed using temperature and solar radiation records from five meteorological stations distributed within the study area. Crop coefficient maps were then developed based on water requirements, identifying crop classes, assigning each class a crop coefficient value. Finally, GIS techniques were used to develop maps representing crop evapotranspiration under standard conditions (crops under optimal conditions), which is the result of the hypothetical evaporation and crop coefficient maps. To estimate crop water demand, a methodology based on the distribution of water content in the soil was used, assuming that, for each month, if the amount of available water resulting from the sum of monthly rainfall and the water stored in the topsoil is sufficient to meet crop needs, irrigation is not necessary. Otherwise, if rainfall is insufficient and soil storage is depleted, there is a deficit that must be supplied by irrigation. The results of this study estimated a demand of 70 Mm$^3$ per year, that is, 100 Mm$^3$ per year of irrigation needs if an average irrigation application efficiency of 70% is considered. Additionally, this approach can be combined with other tools; for example, Alfaro et al. [42] uses the crop water demand-based method to determine groundwater withdrawals. However, their study is based on a groundwater model developed in MODFLOW, from the U.S. Geological Survey, and the main objective of their work is to present a numerical groundwater model that uses a limited (sparse) dataset that is applied to a real case study in a semi-arid region, adopting alternative methods to cope with data sparsity. Its process consists of developing a model based on GWW values using irrigated areas and crop water requirements, subsequently, these extractions serve as input for the WELL package in MODFLOW. The main steps are: (1) estimation of the total irrigated area (temporally and spatially), and (2) determination of crop water requirements that are input to the groundwater model.

On the other hand, Vu et al. [20] conducted an evaluation of groundwater exploitation by developing two approaches, the first one is based on local knowledge through a qualitative field study of groundwater level fluctuations and extractions. This first approach is based on the groundwater balance, which estimates the extraction with the remaining terms of the water balance equation, considering the knowledge of official sources to establish the variables that compose the equation. The second method is based on combined land use data and local knowledge on cultivation and irrigation practices, with this information and using land use and population maps, the GWW is estimated for each of the following groups: irrigation, domestic use and livestock.

*Satellite-based estimation method*: Methods based on teledetection and geographic information systems (GIS) can promote accuracy when estimating the exploitation of an aquifer [56]. For example, Ruud et al. [13] developed a GIS-based water balance model to estimate annual groundwater pumping applied in an agricultural area with a semi-arid climate in the southern San Joaquin Valley, California, USA. GIS-based hydrologic modeling gathers and processes available input data to calculate key components of the basin-scale water balance model, estimating and evaluating groundwater pumping and storage changes. Following the same approach, Ahmad et al. [40] propose a technique for estimating net groundwater use in large irrigated areas by combining teledetection and water balance. This methodology is based on the combined use of teledetection information and GIS techniques to estimate the components of the water balance which are: irrigation rate distribution, net precipitation rate, evapotranspiration rate, and change in soil moisture storage in the unsaturated zone. These can be used to estimate the groundwater net use in agriculture.

In the same way, Castaño et al. [32] present a methodology based on teledetection and GIS for the regulation and quantification of groundwater abstractions in regional aquifers for agricultural use in semi-arid climates. Based on the area used for agriculture and knowledge of the water requirements of each crop, the theoretical amount of water needed for those crops to reach the stage of development visible in the satellite images is calculated. Subsequently, when the area of crops dependent on groundwater collection and agricultural practices is known, a correction coefficient is applied to translate the theoretical amount of water to real values applied to each crop in the area. Finally, all the generated information (distributed in space and time) is integrated into a hydrological information system, which makes it possible to see the relationships between all the water balance elements.

On the other hand, Al-Bakri et al. [35] maintains the approach of Castaño et al. [32] by applying this methodology in Jordan, where was used geospatial techniques for auditing water for irrigation uses. The work was based on the evaluation of GWW records in relation to irrigated areas and estimated water consumption for crops in three river basins: Yarmouk, Amman-Zarqa and Azraq. Therefore, the mapping of irrigated areas and crop water requirements was developed using teledetection data from Landsat 8 satellite and meteorological records with daily periodicity. The methodology relied on visual interpretation and unsupervised classification for remote sensing data. Net (NCWR) and gross (GCWR) crop water requirements were calculated by merging crop evapotranspiration calculated from daily weather records. In addition, groundwater withdrawals records used for irrigation were compared against crop water consumption, assessing whether these records were within the safe yield range, which is known as the pumping rate at which groundwater can be withdrawn without causing a long-term decline in water levels.

Technological advances are relevant in approaches using GIS techniques, as they offer new opportunities for the application of teledetection and geographic information systems. For example, estimating GWW rates by constraining a hydrologic model by integrating a parallel flow simulator called ParFlow coupled to a community land model (CLM). ParFlow is a groundwater flow model that simulates spatially distributed surface and subsurface flow as well as land surface processes including evapotranspiration and snowpack developed as a collaborative effort among several institutions: the Juelich

Research Center, Princeton University, Lawrence Livermore National Laboratory, Colorado School of Mines, the University of Bonn, Washington State University, Syracuse University, and the University of Grenoble Alpes [57,58]. On the other hand, the Common Land Model was developed as a multi-institutional code by a grassroots collaboration of scientists who have an interest in making a general land model (Land Model) [59]. In this approach, water storage changes modeled by ParFlow-CLM are fitted to water storage anomaly estimates from NASA's Gravity Recovery and Climate Experiment (GRACE) satellite and historical water table elevation data by dynamically adjusting groundwater pumping rates; likewise, variable irrigation rates based on soil moisture deficits and crop demand are applied in the model [60,61].

For their part, Parizi et al. [33] use an approach called "representative pumping well network" (RPWN). This methodology is based on the superposition of ten important characteristics—(1) type of alluvial deposits; (2) aquifer thickness; (3) pumping well saturation thickness; (4) pumping flow rate; (5) annual pumping time; (6) pump outlet pipe diameter; (7) transmissivity; (8) type of water consumption; (9) land use/land cover; and (10) distance of the well from main roads. All of these characteristics are applied in a GIS environment that classifies into a series of zones in which their withdrawals are statistically different. In this way, the GIS platform incorporates the hydrogeological characteristics of the aquifer and the properties of the pumping wells to estimate the total GWW of the aquifer by pumping wells.

Forstner and Gleeson [19] use a methodology based on the sectoral method to determine the GWW by different sectors and uses, such as fish aquaculture, agricultural, industrial, domestic and private use. In each of the sectors, a methodology is established based on different criteria that facilitate the estimation of the groundwater volume for each sector, in this way, discharges are attributed to a percentage that may well be population, type of product manufactured by the industry or discharge volumes associated with agriculture. In addition, groundwater use was classified under the following criteria: the first by distribution, either through municipal water systems or through private wells, and the second by the main sectors of groundwater use (domestic, industrial, agriculture and fish aquaculture). Thus, this methodology has been successfully approached using GIS techniques to characterize the areas.

On the other hand, Ray et al. [29] seek to estimate groundwater consumption for irrigation at small scales by studying the spatio-temporal patterns of groundwater consumption and using measurements from 43 wells located in different hydrostratigraphic units. Additionally, surveys were used, which included obtaining information from well/borehole owners on monthly pumping hours and other details related to cultivation practices. As a result, the creation of a small-scale GIS-based database of the number of extraction structures and groundwater consumption was obtained.

*Groundwater modeling approach:* In this approach, the number of pumping wells and the extraction volume are calibrated using a numerical model of the groundwater system [33]. For example, Moreo et al. [36] used a groundwater flow model to estimate GWW in Death Valley, Nevada and California, USA. The extraction locations were estimated using the numerical model. This approach was also applied by Martos-Rosillo et al. [18] in the province of Seville, Spain, using time series analysis and a numerical groundwater model to quantify groundwater exploitation and estimate the mean annual recharge in two carbonate aquifers. Meanwhile, Tsanis and Apostolaki [21] present a method for estimating annual groundwater extraction based on the water balance, considering the discharge of a basin in combination with measurements of the groundwater extracted volume.

On the other hand, Shao et al. [37] used an inversion method of the water balance principle, estimating the GWW by adjusting the simulated groundwater levels with the observed ones based on the concept in which the condition in which groundwater extraction is lower than the actual amount, the simulated groundwater level would be higher than the actual level, and vice versa by establishing an equilibrium equation according to the groundwater extraction in the model. For this purpose, they used a well-calibrated

groundwater model. The principles of the inversion method proposed in this study are based on the water balance and hydrogeological characteristics of the study area. To cope with a large number of pumping wells during the inversion procedure, the study area was divided into pumping subareas. The GWW estimation was raised based on the numerical groundwater model, doing adjustments to groundwater withdrawals based on the inversion method, establishing that the best fit is obtained when the difference between observed and simulated water table levels is lower than those set by the user.

Likewise, Su et al. [34] established a groundwater flow and land subsidence modeling in the flat area of Tianjin, China, where land subsidence is posited as a result of groundwater exploitation. Hydrogeological and geological information, as well as groundwater monitoring data parameters (such as groundwater level and land subsidence value) were used. To better understand the correlation between groundwater exploitation and land subsidence, the groundwater flow simulation program MODFLOW (from the USGS) and the land subsidence simulation package were used. Groundwater flow in the study area was considered as a 3D flow through a porous medium, and land subsidence is considered as a one-dimensional vertical deformation. The annual groundwater exploitation value was estimated by adjusting the calculated water table with the observed water table.

Lin et al. [27] employ the soil and water assessment tool known as SWAT (Soil and Water Assessment Tool) as well as MODFLOW (USGS) to separately run and acquire certain hydrological components such as recharge and storage change. Subsequently, it uses the water balance method to estimate pumping rates with these components. In this methodology SWAT is used to accurately estimate vertical recharge and identify potential recharge zones based on the physical characteristics of the watershed. Subsequently, storage and net infiltration are estimated with MODFLOW according to hydrogeological characteristics, and finally the water balance method is applied to estimate pumping rates.

Liu et al. [28] developed a method for the characterization and quantification of regional groundwater pumping using pumping source identification that combines signal analysis with a groundwater flow simulation model. The method focuses on two parts, the first uses independent component analysis (ICA) to identify the main pumping types and variations in groundwater elevations that generate these signals. This relates a signal to each type of pumping, either for agricultural or industrial use. Second, the groundwater model is used to estimate and quantify the amount of pumping for each type of signal, matching the simulation to variations in the water table.

*Water table fluctuation method (WTF):* This approach uses the relationship between the change in groundwater storage with fluctuations in water tables (specific yield in the case of unconfined aquifers), which is based on the assumption that the increase in the water table at an observation point during the recharge season is caused by recharge through the water table, and such an increase is multiplied by the specific yield to obtain a direct estimate of recharge [21]. Usually, the water table fluctuation method is combined with the groundwater balance equation and a geostatistical method to estimate the annual GWW pumping [33]. For example, Martínez-Santos and Martínez-Alfaro [31] developed an adaptation of the WTF method to estimate groundwater pumping in agricultural areas of central Spain, coupling the water table fluctuation method with the groundwater equilibrium equation. Similarly, Yang et al. [22] developed a modified WTF method to quantitatively characterize regional groundwater discharge patterns in aquifers under stress caused by intensive agricultural pumping. The study develops the method called water table fluctuation regression (WTFR) and is designed to characterize systems that are driven by both precipitation recharge and net discharge processes, as the basis of the method the water table hydrograph at an observation point is defined by two parameters: infiltration efficiency and discharge modulus. The former is a relationship between the amount of precipitation and the increase in the water table and the latter is a 12-element matrix representing the net discharge pattern, defining the discharge modulus as the net head reduction due to the collective result of pumping discharge and irrigation return.

*Artificial intelligence-based methods*: The necessity to address groundwater problems using alternative techniques that can be relatively simpler and less expensive has led researchers in different parts of the world to explore machine learning (ML)-based models [62]. Machine learning methods have been widely used in recent years in many fields of water resources (e.g., [63–65]). Some of the models based on artificial intelligence and machine learning techniques are artificial neural networks (ANNs) and adaptive neuro-fuzzy inference systems (ANFIS), which have been developed recently and have demonstrated their effectiveness in hydrological system applications [66,67]. Therefore, the application of artificial intelligence-based methods for GWW determination is a tool that authors resort to; for example, Majumdar et al. [44] approaches the correlations between various water balance measurements and groundwater withdrawals in a machine learning framework, which learns the relationship between different datasets and uses them predictively. The machine learning approach to predict local-scale groundwater withdrawals uses raster and vector files with 5 km spatial resolution, taking information from various satellite, e.g., evapotranspiration (MODIS), precipitation (PRISM), and land use data (USDA-NASS), which are related in different ways to groundwater withdrawals. On the other hand, Li et al. [25] developed an artificial neural network (ANN) model based on the back-propagation algorithm to estimate groundwater discharge. For this purpose, water table, precipitation and groundwater discharge were set as inputs to the ANNs, using 5 hidden layers and 144 data from the groundwater discharge records to train the ANNs. The results are compared with official data and water balance method.

## 4. Discussions

Establishing an optimal method to quantify groundwater discharge is complicated, thus it could be inferred that direct methods are the oldest and most reliable; however, this is not always the case due to the uncertainty they may present at the time of taking measurements, which could be attributed to human error or failures in the measurement systems themselves, temporary abatement cones, occasional or emergency uses, etc. In addition, the difficulty and high costs at the time of its implementation mean that it becomes a complicated method to execute in areas that have a large number of extraction wells, whose extension and intensive agricultural use means that these methods are not very applicable or when the users have different interests, such as private wells, federal wells and when the sectors are different, such as industrial, agricultural or urban. These are some of the major challenges faced by direct measurements. Some applications face this problem by using electrical measurement records, statistical methods and methodologies based on the participation of farmers, trying to establish greater reliability in direct measurement records, since an advantage of these methods is the security they can provide to obtain accurate measurements, which in some cases serve as a starting point for the development of an indirect method.

On the other hand, indirect methods are a relatively inexpensive solution to address some of the limitations involved in direct methods; however, there are certain considerations among the five groups described above.

Estimation method based on crop water requirements: Among the main characteristics of this method is the possibility of using the vegetation cover of a given area together with the estimation of crop water demands, allowing the estimation of groundwater pumping based on the crop type and its water demands. This approach shares a strong link with advances in teledetection techniques and geographic information systems, allowing spatial and temporal tracking of vegetation cover. However, on its own it has some drawbacks, since crop water demands can vary due to climatic conditions such as rainfall or temperature, and it also involves determining and monitoring agricultural practices, which can be a challenge. Nevertheless, this method is an interesting alternative if the study area is mainly dominated by agriculture.

Methods based on groundwater flow modeling have some advantages since they use groundwater monitoring and time series data, which allows dynamic monitoring

to estimate groundwater exploitation, and are models used to understand an aquifer system, characterize groundwater flow and simulate different management scenarios. A disadvantage of the groundwater modeling approach is the need to know all the elements that describe the aquifer in terms of time and space, and the difficulty increases when the number of pumping wells is large. In addition, the limitation of human resources, materials and technological methods makes it difficult to obtain sufficient and important data—for example, the complexity of the aquifer, such as the presence of clay lenses, the presence of confined or semi-confined aquifers, carbonate aquifers, outcrops, subsidence and compaction. This makes it possible for this approach to be used in areas with a reduced number of extraction wells, and it has a good potential for adaptation with other tools or methods.

The WTF method is mainly designed for natural systems. It has important limitations since it does not take into account water table fluctuations resulting from pumping; therefore, it receives adaptations to determine water withdrawals. In some cases, adaptations of this approach aggregate discharge signals; however, they may not consider some important aspects such as evapotranspiration as a significant factor for water table fluctuations if the aquifer is shallow. This approach benefits when applied for short periods in regions with strong water table fluctuations.

The estimation method based on satellite data is usually a feasible solution that relies on modern tools such as satellite information, which makes it possible to evaluate land and water resources, and the coupling with teledetection methods and geographic information systems is a very good combination, since it is a cost-effective way to obtain large-scale information. However, it may present some difficulties due to the integration of the different spatial and temporal resolutions that can be obtained from satellite data. This approach has a great applicability in large scale areas.

Methods based on artificial intelligence allow the implementation of technological advances in data analysis in a simple way, taking advantage of the correlation between the various datasets that can be both water balance and groundwater withdrawals, creating a learning framework, in addition, it can be used in a predictive way; however, its great disadvantage is the availability of data. Sometimes it is not possible to obtain updated information or there are important gaps in the time series; this could be a disadvantage if the training of the model depends on historical data. This approach has great versatility to be adapted with other tools.

## 5. Conclusions

A review of direct and indirect methods has been carried out, and the results are summarized and organized in Table 1 so that the trends in recent years can be identified., with partial and general results, which can provide guidelines applicable to researchers who want to perform similar work in this field. Likewise, the different methods for groundwater pumping estimation that have been reviewed in this paper have advantages and disadvantages in their application. Direct approaches are very old and still work; the uncertainty in their application comes from possible human errors or in the measurement system, in addition to the irrigation tendencies by the farmers that also play an important role in the application of direct approaches. Such methods were widely used in the 1970s; however, it is very costly to implement measurement devices in areas with a large number of extraction wells. On the other hand, indirect methods have evolved along with technological advances. The first indirect approaches were based on crop water demands, with remote sensing data and satellite information playing a very important role, which have brought significant improvements and precision to this approach. Additionally, numerical models are of great importance in the estimation of groundwater pumping based on the principle of water balance, the structure of aquifer systems, model boundaries, recharge and discharge conditions. The trend in recent years of this approach is inclined toward coupling with other techniques. The adaptation of the WTF method to estimate groundwater pumping is still available in a few studies; however, it has demonstrated

its applicability to assess the estimation of groundwater pumping. For their part, the methods based on teledetection and GIS for groundwater pumping estimation are those that have shown a significant increase in their applications, with the trend in recent years, via technological advances, making application easy in large areas with access to satellite databases. Finally, artificial intelligence-based approaches have proven to establish a learning framework applicable in the field of groundwater pumping estimation; however, they are just beginning to be applied. Therefore, defining an appropriate model will depend on the purpose of the model and also on other factors such as data availability and computational power.

**Author Contributions:** M.A.M.-G.: conceptualization; M.A.M.-G.: investigation; M.A.M.-G., J.R.C.-G., J.R.-H., C.S.H.-O., J.J.V.-L. and A.F.-N.: data organization; M.A.M.-G.: writing—original draft; M.A.M.-G., J.R.C.-G., J.R.-H., C.S.H.-O., J.J.V.-L. and A.F.-N.: writing—review and editing; J.R.-H., C.S.H.-O., J.J.V.-L. and A.F.-N.: supervision; J.R.C.-G.: project administration. All authors have read and agreed to the published version of the manuscript.

**Funding:** Authors appreciate the doctoral scholarship granted to Marco Antonio Meza Gastelum from Mexican National Council for Science and Technology (CONACYT) grant number 4.5 UMA.

**Conflicts of Interest:** The authors state that they have no conflict of interest to declare.

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
