# Peer review of "Review of Groundwater Withdrawal Estimation Methods"

_water, doi:10.3390/w14172762_

Round 1

Reviewer 1 Report

A study is presented titled Review of Groundwater Pumping Estimation Methods. The authors have strong knowledge about the theoretical science and addressed hot topic. The topic is very interesting for hydrogeology.

Abstract

This section is not properly addressed in scientific way. It is suggested to add conclusion of your research properly in this section.

Introduction

In the introduction, the section needs revision to add the significance and your achievement of the paper.

Figure 1. PRISMA flow chart. Adapted from [1].

Comments: It is suggested write something how it will helpful for your work.

Figure 2. Annual publications of methods for estimating groundwater pumping.

It is suggested to separate data of under developing and developed continues.

Artificial intelligence-based methods

It is suggested to enhance this section with some examples of ML and deep learning.  

Reviewer 2 Report

This paper deals with a very interesting and urgent topic, but, unfortunately, it needs to be deeply revised, starting from the title, where it is referred to Groundwater Pumping Estimation Methods, while in the test it is usually referred to the term groundwater withdrawal, which is definitely more appropriate to the topic, described in the paper. On the other hand in the review they are presented many different methods, without any relationship to the scale of the study they have applied to. In such a studies the extension of the area under study is very important to find out the reliability of the applied methods. Moreover, the analysis of the methods in the literature is not so esaustive  as it missed, for instance, the attached one, which can be referred to the indirect class. Finally, I think the paper needs to be reviewed starting from the classification of considered methods, which have to be better censored in literature, and divided according to the type, the scale of the study area, and their being direct or indirect. After this renewing the paper can be reconsidered for publication

Round 2

Reviewer 2 Report

I noticed that my suggestions have been correctly included in the paper. So I suggest only few little revisions of english language, and in this version the paper can be published